# Low-Cycle Fatigue Behaviour of Titanium-Aluminium-Based Intermetallic Alloys: A Short Review

**John J. M. Ellard** [1], **Maria N. Mathabathe** [2], **Charles W. Siyasiya** [1] and **Amogelang S. Bolokang** [2,3,4,*]

1. Department of Material Science and Metallurgical Engineering, Faculty of Engineering, Built Environment and Information Technology, University of Pretoria, Pretoria 0028, South Africa; u20809515@tuks.co.za (J.J.M.E.); charles.siyasiya@up.ac.za (C.W.S.)
2. Council of Scientific Industrial Research, Manufacturing Cluster, Advanced Materials Engineering, Meiring Naude Road, Pretoria 0001, South Africa; nmathabathe@csir.co.za
3. Department of Physics, University of the Free State, Bloemfontein 9300, South Africa
4. Department of Physics, University of the Western Cape, Private Bag X 17, Bellville 7535, South Africa
* Correspondence: sbolokang@csir.co.za

**Abstract:** Over the past decade, relentless efforts have brought lightweight high-temperature $\gamma$-TiAl-based intermetallic alloys into real commercialisation. The materials have found their place in General Electric's (GE) high bypass turbofan aircraft engines for the Boeing 787 as well as in the PW1100GTF engines for low-pressure turbine (LPT) blades. In service, the alloys are required to withstand hostile environments dominated by cyclic stresses or strains. Therefore, to enhance the fatigue resistance of the alloys, a clear understanding of the alloys' response to fatigue loading is pivotal. In the present review, a detailed discussion about the low-cycle fatigue (LCF) behaviour of $\gamma$-TiAl-based alloys in terms of crack initiation, propagation and fracture mechanisms, and the influence of temperature and environment on cyclic deformation mechanisms and the resulting fatigue life has been presented. Furthermore, a comprehensive discussion about modelling and prediction of the fatigue property of these alloys with regard to the initiation and propagation lives as well as the total fatigue life has been provided. Moreover, effective methods of optimising the microstructures of $\gamma$-TiAl-based alloys to ensure improved LCF behaviour have been elucidated.

**Keywords:** low-cycle fatigue; fatigue mechanisms; fatigue property models; $\gamma$-TiAl microstructures; alloying elements influence; $\gamma$-TiAl alloys; $\gamma$-TiAl properties



## 1. Introduction

### 1.1. Binary Phase Diagram of TiAl Alloys

Although titanium can be substitutionally or interstitially alloyed with elements such as B, C, Mo, V, Zr, Si, Cu, Nb, Sn and Cr to produce metallurgical structure classified titanium alloys, viz. near-$\alpha$, $\alpha$-$\beta$ and metastable-$\beta$ alloys [1], the most widely used alloying element in titanium alloys is aluminium. Al is the only common metal raising the transition temperature and has large solubilities in both $\alpha$ and $\beta$ phases. Moreover, it strengths the $\alpha$-phase up to a temperature of 550 °C and has a low density of 2.7 g/cm$^3$ [2]. The widely used TiAl binary phase diagram for alloys with compositions near TiAl is shown in Figure 1.

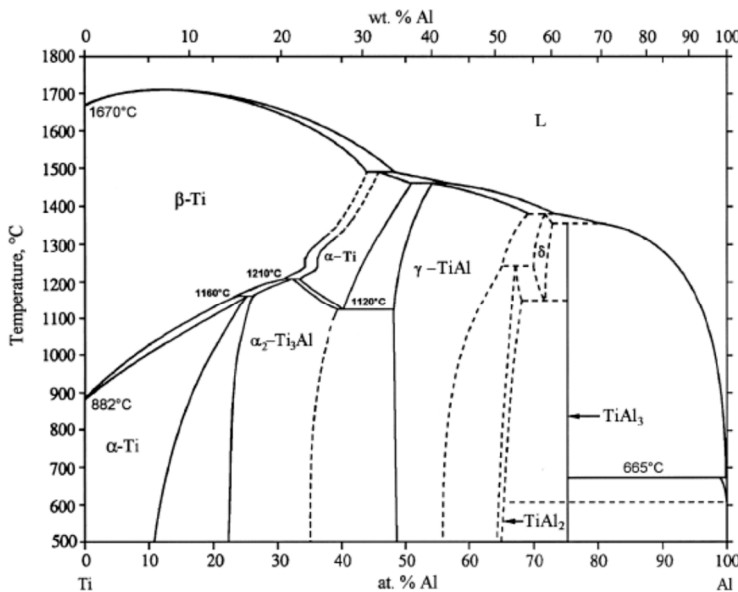

**Figure 1.** TiAl binary phase diagram [3].

Illarionov et al. [4] elucidated that the $\beta$ phase can be stabilised by using two categories of alloying elements, viz. $\beta$ isomorphous elements such as vanadium, molybdenum, tantalum and niobium, and $\beta$ eutectoid forming elements such as chromium, iron, manganese, nickel and silicon. When enough contents of these elements are added to the TiAl alloys, the $\beta$ phase becomes stable at room temperature. Subsequently, the $\beta$-phase acts as a ductile second phase due to its 12 independent slipping systems which can improve the deformation behaviour and arrest or hinder the growth of micro-cracks that form in the brittle major constituents, the $\alpha_2$ and $\gamma$ phases [4]. The Ti-Al phase diagram gives conventional titanium alloys, and more importantly, forms the basis of the recently developed, implemented and commercialised lightweight high-temperature titanium-aluminides. Analysing critically the Ti-Al binary phase diagram shown in Figure 1, one could notice the existence of three compounds in this system, viz.: $Ti_3Al$ ($\alpha_2$), TiAl ($\gamma$), and $TiAl_3$. However, for many years, substantial research work has been dedicated only to $\alpha_2$-$Ti_3Al$ and $\gamma$-TiAl compounds. The $\alpha_2$-phase has the crystal structure of $DO_{19}$ (ordered hexagonal) while the $\gamma$-phase has $L1_0$ (ordered face-centred tetragonal) structure as shown in Figure 2 [5]. In the $\gamma$-phase, for the stoichiometric compound, c/a = 1.015; the tetragonality ratio increases to 1.03 with increasing aluminium concentration and decreases to 1.01 with decreasing aluminium. There exists the crystallographic orientation relationship of (111)∥(0001) and [110]∥[11$\bar{2}$0] with a semi-coherent lattice match when the $\gamma$-phase is precipitated out of the $\alpha_2$ matrix [6–8].

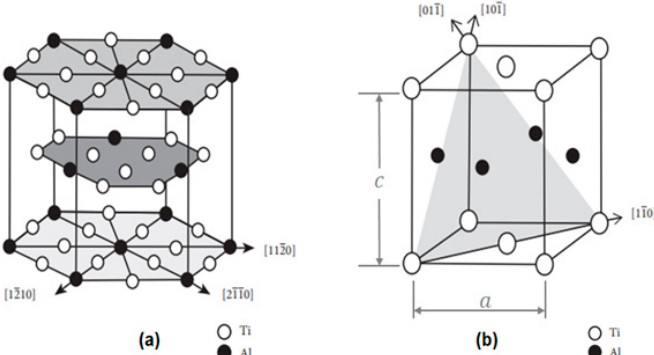

**Figure 2.** Crystal structures of TiAl compounds: (**a**) hexagonal $\alpha_2$-$Ti_3Al$ phase ($DO_{19}$); (**b**) tetragonal $\gamma$-TiAl phase ($L1_0$) [5].

Further examination of the middle section of the Ti-Al phase diagram as depicted in Figure 3, three solid phases, viz. $\alpha_2$-Ti$_3$Al, $\gamma$-TiAl and the high-temperature $\alpha$-Ti, and two-phase reactions, viz. peritectic $L + \alpha \rightarrow \gamma$ reaction and eutectoid $\alpha \rightarrow \alpha_2 + \gamma$ reaction, exist [9]. Although the single-phase alloys offer excellent resistance to environmental attack (oxidation and hydrogen absorption) and initially drew the interest of several researchers, their uttermost lack of ductility and fracture toughness at room temperature, particularly in the binary state rendered them irrelevant for any engineering application [10]. Nevertheless, extensive research has shown that the two-phase ($\alpha_2$-Ti$_3$Al + $\gamma$-TiAl) or multi-phase $\gamma$-TiAl based alloys that solidify through the $\beta$-phase (i.e., $L + \beta \rightarrow \beta$) and the $\alpha$-phase (i.e., peritectic reaction: $L + \beta \rightarrow \alpha$) with Al composition between 40 and 48 at.% [Ti(40–48)] have engineering significance [11,12]. These alloys are known as $\gamma$-TiAl-based alloys with the $\gamma$-(TiAl) and $\alpha_2$-Ti$_3$Al being the primary and secondary phases, respectively. Research has shown that for best properties, the alloys should contain more than 90 and 5 volume percentage of $\gamma$ and $\alpha_2$ phases, respectively [13,14], and the alloys can be categorised as either $\beta$ or peritectic solidifying $\gamma$-TiAl-based alloys depending on the alloy composition [15].

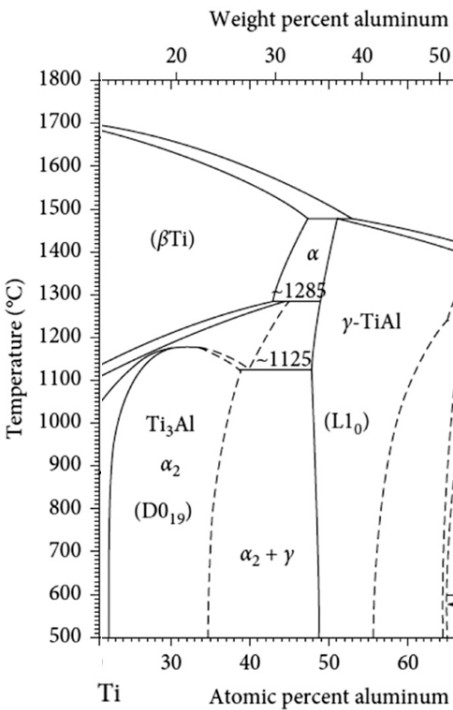

**Figure 3.** A central portion of the Ti-Al phase diagram [9].

### 1.2. Microstructures and Properties of TiAl Alloys

Appel et al. [16] stated that on either cooling the $\gamma$-TiAl-based alloys from the high-temperature zones or on subsequent heat treatment, a diversity of distinct phase transformations can take place. In general, one could obtain four types of microstructures in $\gamma$-TiAl-based alloys, viz. near $\gamma$, near lamellar, fully lamellar and duplex [17,18]. The near $\gamma$ microstructure consists of equiaxed $\gamma$ grains and $\alpha_2$ particles at the grain boundaries and triple points whereas the near lamellar comprises $\alpha_2 + \gamma$ lamellae and $\gamma$ grains with the lamellae being larger than the $\gamma$ grains. However, Kim [14] stated that for wrought-processed materials, these microstructures can generally be represented by the fine wrought duplex (DP) and coarse fully lamellar (FL) microstructures as summarised in Table 1 together with their mechanical properties [15,19–23].

**Table 1.** Microstructures and mechanical properties of γ-TiAl-based alloys.

| Microstructure | Characteristics | Mechanical Properties |
|---|---|---|
| Fully lamellar (FL) | Coarse lamellar $(\alpha_2 + \gamma)$ grains | <ul><li>Superior creep resistance</li><li>Better fracture toughness</li><li>Good fatigue resistance</li><li>Better room and elevated temperature strength values (applicable for fine grain size)</li><li>Much lower ductility values</li></ul> |
| Duplex (DP) | Fine equiaxed γ-grains and $(\gamma + \alpha_2)$ lamellae | <ul><li>Better strength and ductility</li><li>Poor creep resistance</li><li>Inferior crack propagation resistance</li><li>Longer total fatigue life</li></ul> |

In general, the TiAl alloys lack ductility, especially at room temperature. Wu [24] pointed out that the ambient temperature ductility was inadequate, and was particularly about 1% for all TiAl alloys. This property (low ductility) shown in Figure 4 was a major drawback, especially for structural components, since 1% elongation is commonly an accepted minimum value, and cast components in particular rarely reach this level [24].

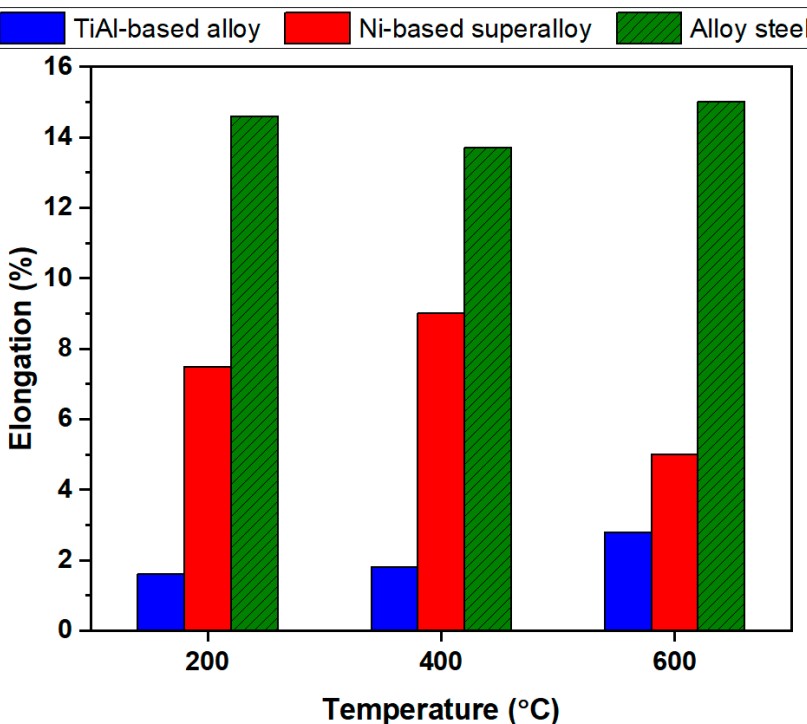

**Figure 4.** Ductility as a function of temperature for a TiAl-based alloy, a NiAl-based alloy and alloy steel.

As shown by research, the mechanical properties of TiAl alloys such as ductility are majorly influenced by the composition of the alloy as well as its microstructure [25]. However, Al content appears to play a more significant role in controlling the amount of the resulting microstructural constituents. As demonstrated by Huang et al. [26] in their investigation of the influence of Al content in the range of 43–55 (at.%) on TiAl alloys' room temperature ductility, the two-phase composition of Ti-48 Al (at.%) exhibited the maximum elongation owing to the optimum volume ratio (3–15%) of $\alpha_2/\gamma$. At this composition, the amount of γ-phase was high enough to accommodate the majority of plastic deformation [27]. Above this optimum Al content, the volume fraction of $\alpha_2$-phase

decreased. However, grain growth became pronounced when heat-treated in the $\alpha + \gamma$ phase field, and in the single-phase region, the increase in the tetragonality ratio of the $\gamma$-phase decreased the ductility. When the Al content was below the optimum, the ductility decreased. This was attributed to the increase in the brittle $\alpha_2$-phase which eliminated the beneficial effect of refined microstructure. In terms of the lamellar thickness of $\gamma$ plates in the binary Ti-Al alloys, simulation results obtained by Zhang et al. [28] in the Al content range of 40–43 (at% Al) indicated that the thickness varied with Al content by decreasing and then increasing as the Al content was increased. The optimal composition was found to be the one with Al concentration of 41.5%.

### 1.3. Influence of Alloying Elements on TiAl Alloys

However, significant progress in the development of $\gamma$-TiAl alloys has been made in the past decades. Several mechanical properties of the alloys such as strength, fracture toughness, ductility, creep, oxidation and fatigue have been improved by mainly the use of alloying elements. The most common alloying addition for the $\gamma$-TiAl-based alloys is Nb, but other elements also have been added to the alloys with a clear benefit, at least to selected properties, as illustrated in Table 2. The principle beneficial effects of alloying as elucidated by several researchers include relaxing the slip modes restriction, holding back the kinetics of ordering or altering the degree of long-range order, introducing the disordered $\beta$ phase or the ordered $\beta$ phase in the alloys thereby altering their constitution to be able to arrest or hinder the growth of micro-cracks that form in the brittle major constituents, and permitting microstructural control and refinement by adjusting the transformation behaviour during heat treatment and processing [29,30].

**Table 2.** Effects of alloying elements on properties of $\gamma$-TiAl-based alloys.

| Alloying Element | Effects |
|---|---|
| Nb | • Improves strength and toughness [30,31]<br>• Improves creep and oxidation resistance [32,33]<br>• Stabilises $\beta$-phase to promote hot-workability [34,35] |
| Mn | • Increases room temperature ductility [30,36]<br>• Improves ductility and plasticity through grain refinement [30,37] |
| Cr | • Reduces the oxidation rate at 800–900 °C and 1000–1100 °C [38]<br>• Stabilises B2 [36] |
| V | • Increases ductility [30]<br>• Stabilises $\beta$-phase [39] |
| Mo | • Improves oxidation resistance [29,30,40]<br>• Stabilises $\beta$-phase [41–45]<br>• Strengthens alloy matrix [46,47]<br>• Increases creep resistance by delaying diffusion [48]<br>• Improves compression properties at high temperatures [49] |
| B | • Refines grains [50,51]<br>• Provides good castability [16]<br>• Increases ductility [52] |
| C | • Improves fatigue and creep resistance [15,29,53]<br>• Refines the lamellar spacing [54,55]<br>• Improves strength [53] |
| Si | • Refines microstructure [30]<br>• Improves creep strength [56,57]<br>• Increases oxidation and corrosion resistance [58,59]<br>• Stabilises the microstructure [60,61]<br>• Improves fracture toughness and flexural strength [62] |

| Alloying Element | Effects |
|---|---|
| Ta | • Improves oxidation resistance [30,63]<br>• Enhances creep resistance [64]<br>• Refines lamellar colony sizes [65] |
| Zr | • Refines microstructure [66]<br>• Improves compressive strength [67]<br>• Enhances fracture strain and yield stress at intermediate temperatures [68]<br>• Enhances creep resistance [68] |
| Sn | • Increases oxidation resistance [69]<br>• Refines microstructure [70,71]<br>• Improves liquidity [70] |
| Fe | • Stabilises β-phase [29,72]<br>• Improves liquidity [72]<br>• Reduces lattice tetragonality [72]<br>• Refines grains and enhances strength [72] |
| Y | • Promotes grain refinement [67]<br>• Increases strength and elongation [67]<br>• Increases hot-deformability [67] |

These alloying elements and the corresponding attractive properties they induce to the γ-TiAl alloys have led to the development of several alloy systems as shown in Table 3.

**Table 3.** Categories of γ-TiAl-based alloys.

| Generation | Alloy Designation | Series | Ref. |
|---|---|---|---|
| 1st | Ti-48Al-1V-(0.1Wt%) C | Ti-48Al-1V-(0.1Wt%) C | [73] |
| 2nd | Ti-(45–48)Al-(1–3)X-(2–5)Y-(<1)Z<br>X = Cr, Mn, V; Y = Nb, Ta, W,<br>Mo; Z = Si, B, C | 4822 (Ti-48Al-2Nb-2Cr)<br>45XD (Ti-45Al-2Mn-2Nb-0.8vol%TiB2<br>47WSi (Ti-47Al-2W-0.5Si) | [74] |
| 3rd and 4th | Ti-(42–48)Al-(0–10)X-(0–3)Y-<br>(0–1)Z-(0–0.5)RE<br>X = Cr, Mn, Nb, Ta; Y = Mo, W,<br>Hf, Zr; Z = C, B, Si; RE = rare<br>earth elements | BMBF3 (Ti-47.5Al-5.5Nb-<br>0.5W<br>K5 (Ti-46Al-3Nb-2Cr-<br>0.2W-xB-yC/Si)<br>TNB-V5 (Ti-45Al-5Nb-0.2B-0.2C<br>TNM (Ti-43Al-4Nb-1Mo-0.1B)<br>Ti-44Al-4Nb-4Hf-0.1Si-0.1B<br>Ti-44Al-6Nb-1Mo-0.2Y-0.1B | [75–79] |

### 1.4. Applications of TiAl Alloys

According to Duan et al. [80], the first generation of γ-TiAl-based alloys possessed a good combination of ductility and creep resistance. Nevertheless, the values of the properties were inadequate to meet the requirements of any engine components. Therefore, this led to the development of the second generation of the alloys, which was found to possess better mechanical properties, viz. room temperature plasticity, high-temperature strength, creep and fatigue, than the first generation. The alloys were commercially implemented in General Electric's (GE) high bypass turbofan aircraft engines for the Boeing 787 [3,80]. However, there was a need to broaden its application range, to improve its high-temperature capabilities and increase the window for hot working. Therefore, the third generation was subsequently developed and found its usage in the PW1100GTF engines for low-pressure turbine (LPT) blades [3,81].

As one could notice, many of the applications identified for γ-TiAl alloys, such as gas turbine blades in jet engines and turbo rotors in automobiles, involve rotation of the components at elevated temperatures (up to about 800 °C) [82] while carrying a specific

amount of load. The components are also subjected to large thermal expansions and contractions so that they undergo large cyclic strains during a heating and cooling cycle. Viswanathan and Stringer [83] and Hyde et al. [84] elucidated that the main mechanisms of failure of components subjected to high temperatures include creep, fatigue, creep-fatigue, and thermal fatigue. However, research has shown that fatigue is a cause of approximately 90% of all service failures of metal parts. Moreover, Chan et al. [85] reported that the addition of several elements in an alloy brings about complex microstructures in the alloy, which could result in some features that could be potential sites where competing fatigue crack initiation mechanisms may occur. These sites may include areas of induced plastic constraint, pores and inclusions. In the literature, several reviews of the fatigue behaviour of $\gamma$-TiAl-based alloys are available [86,87]. Hénaff and Gloanec [86] reviewed the fatigue life, cyclic stress/strain behaviour and fatigue crack growth resistance as well as the influence of microstructure, defects, temperature and environment on the fatigue properties. On the other hand, Edwards [87] assessed the progress in the understanding of the high cycle fatigue (HCF) behaviour of $\gamma$-TiAl alloys. However, in these reviews, the fatigue fracture mechanisms, the assessment of the fatigue performance prediction models as well as the thermo-mechanical processing routes that aim at optimising the alloys' microstructures for improved fatigue properties were not incorporated. Moreover, since the large amount of load that the $\gamma$-TiAl components carry during operation contributes significantly to the large cyclic strain levels in the material which results in the number of cycles to failure being relatively small within the region of low-cycle fatigue, it is important to get a deeper understanding of the low-cycle fatigue behaviour of these engineering materials. Therefore, this review focuses on the low-cycle fatigue (LCF) behaviour of $\gamma$-TiAl-based alloys at room and elevated temperatures, some of the fatigue property prediction models that have been applied to $\gamma$-TiAl-based alloys and microstructural optimisation processes for improved LCF property.

## 2. Review Methodology

To discuss the microstructural mechanisms in $\gamma$-TiAl-based alloys subjected to low-cycle fatigue loading, models employed to predict LCF behaviour and processes that aim at optimising the microstructure for improved LCF property in the alloys, gamma titanium aluminide alloys textbooks were consulted to identify the existing gaps. Moreover, previous reviews on fatigue of $\gamma$-TiAl alloys were also studied to be acquainted with what had already been discussed. However, studies on high-cycle fatigue of the alloys were excluded. Furthermore, most of the articles which discussed the deformation mechanisms of the alloys under LCF loading were also excluded because the information was already compiled in one of the gamma titanium aluminides textbooks. Nevertheless, a few articles on the aforementioned subject were used to only provide a detailed discussion on the microstructural mechanisms concerning how a fatigue crack nucleates and propagates in the alloys when subjected to a strain-controlled process, and also the nature of their fracture surfaces which was still not clear in the existing textbooks and reviews. In addition, studies on the modelling of LCF behaviour of the alloys, and their microstructural optimisation processes also played a central role in the present review.

Online databases of Science Direct, Springer, Scopus, Web of Science, PubMed Central and Wiley Online Library were searched for relevant articles. The following keywords were used: (("gamma titanium aluminides" OR "fatigue of gamma titanium aluminides) AND ("mechanisms" OR "low-cycle fatigue" OR "modelling of fatigue behaviour" OR "microstructural optimisation")). Only English language articles published between 2015 and 2023 were preferred. However, due to the limited number of recent articles addressing the low-cycle fatigue of $\gamma$-TiAl-based alloys, earlier published articles were also used. 43 articles matched these criteria as shown in Figure 5. The rest of the references cited in this review were for the background discussion.

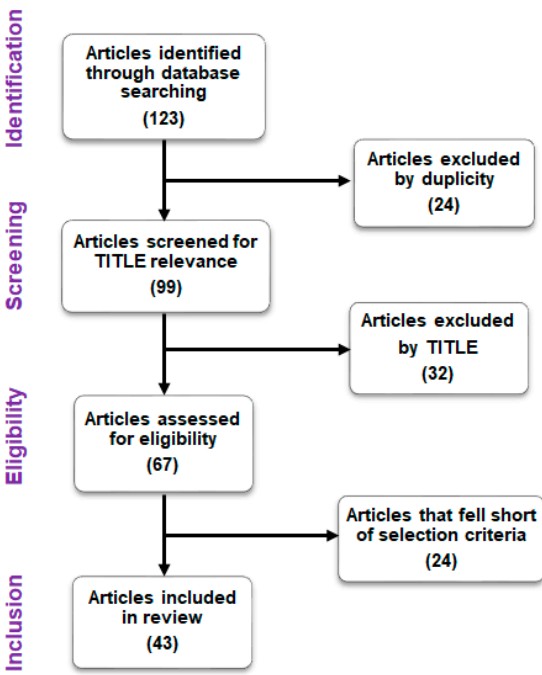

**Figure 5.** Search strategy.

## 3. Low-Cycle Fatigue Behaviour of γ-TiAl-Based Intermetallic Alloys

### 3.1. Microstructural Mechanisms

The fatigue crack initiation, propagation and fracture stages and their corresponding mechanisms in γ-TiAl alloys in the low-cycle regime depend on the type of macrostructure. Recina and Karlsson [88] studied the initiation of cracks at low cycle fatigue of Ti-48Al-2Cr-2Nb (at.%) cast alloy at 600 °C with a duplex microstructure. The specimens were prepared according to American Standards for Testing and Materials (ASTM) E 606–92 with a gauge diameter of 6.35 mm. The researchers observed that fatigue crack initiation occurred on the damaged surfaces, at pores and oxide inclusions just like in any other metallic materials. However, apart from these weak spots, other locations applicable to duplex γ-TiAl-based alloys, viz. debonded γ-grains, grain clusters and larger single-phase γ-grains acted as stress raisers during cyclic deformation. In these sites, the formation of intrusions and extrusions as shown in Figure 6 was observed. Furthermore, they reported that the lamellar colonies of the duplex microstructure were not susceptible to fatigue crack initiation [88].

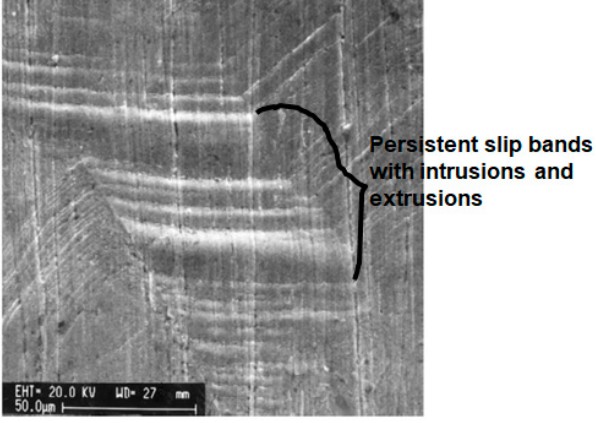

**Figure 6.** SEM micrograph showing extrusions and intrusions in a large γ-grain of duplex microstructure. The micrograph was reproduced with permission from Elsevier [82].

In a different study [82] conducted by the same researchers but working on a different alloy (Ti-48Al-2W-0.5Si (at.%)), a low cycle fatigue test at 600 °C showed that fatigue crack initiation was predominant mainly in large $\gamma$-grains or internally in interdendritic $\gamma$-areas as shown in Figure 7a. In terms of fatigue crack propagation in duplex microstructure, Recina et al. [89] stated that there was some general stable fatigue crack growth as evidenced by the presence of striations shown in Figure 7b. This was because the sizes of the lamellae and $\gamma$-grains were smaller than the critical size of the propagating crack.

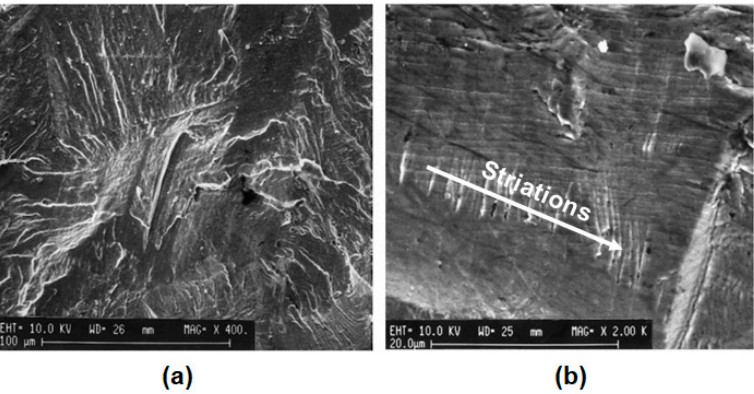

**Figure 7.** SEM micrographs showing (**a**) interdendritic initiation of fracture at a lamellar lath in the duplex microstructure, and (**b**) fatigue striations in the duplex microstructure with spacing of 0.5–3 μm. The micrographs were reproduced with permission from Elsevier [82].

In fully lamellar and near lamellar microstructures, apart from the fatigue crack nucleating near weak spots such as pores, cavities, damaged surfaces and oxide inclusions, Recina et al. [89] observed that the lamellar colonies whose laths oriented perpendicular to the load axis acted as defects in the structure because it was easier for fatigue cracks to nucleate in the interlamellar regions than it was within the lamellar laths. After the occurrence of interlamellar crack initiation, the crack rapidly propagated and caused a catastrophic failure [89]. Furthermore, the researchers pointed out that although the orientation of lamellar lath is random in the NL material, the interlamellar fatigue crack initiation was facilitated by the large colonies unfavourably oriented. In general, when the lamellar structure consists of lamellae whose sizes are larger than the expected critical crack size, their anisotropic properties tend to control fatigue failure [82]. For a notched specimen, Min et al. [90] reported that small fatigue cracks in a nearly lamellar high-Nb TiAl alloy at 750 °C initiated at the central portion of the notch before shifting to the edge portion. Subsequently, the cracks joined together to form a main fatigue crack with an increasing number of cycles. At room temperature, Petrenec et al. [91] observed a cyclically strained surface of a Ti-48Al-2Nb-2Cr-0.82B (at.%) alloy with a nearly lamellar microstructure which indicated cyclic slip localisation. The continued number of cycles generated the persistent slip markings along the $\gamma/\alpha_2$ and $\gamma/\gamma$ interlamellar interfaces in the interior of grains which became the nucleation sites for fatigue cracks [92].

Several researchers have investigated the low-cycle fatigue crack propagation mechanisms of $\gamma$-TiAl alloys with lamellar microstructures at both room and elevated temperatures [93–95]. All the research attests to the fact that fatigue crack propagation follows the interlamellar cracking for the lamellae oriented along the crack direction and the translamellar cracking for the lamellar colonies oriented at an angle with the direction of the growing crack. However, Trant et al. [94] reported that when the estimated size of the crack tip plastic zone exceeded the colony size, a change from trans- to mixed trans-, inter- and intra-lamellar cracking occurred. In general, the rate of fatigue crack propagation in lamellar microstructures depends on the lamellae orientation at the crack tip where the majority of the cyclic strains concentrate [96]. When the fatigue crack is parallel to the lamellar laths, rapid and accelerated crack propagation is expected, while when the fatigue

crack is perpendicular to the lamellar laths, steady and accelerated propagation is likely to take place, with no rapid propagation [90].

Concerning fracture mechanisms, Malakondaiah and Nicholas [97] observed a mixed mode comprising delamination, translamellar fracture and stepwise fracture in the FL microstructure of Ti-46Al-2Nb-2Cr alloy fatigue tested at 650 and 800 °C with the specimen dimensions of 19.05 mm gauge length and 6.35 mm diameter with uniform end connections of 12.7 mm diameter (ASTM E 606–92). At room temperature in the air, Dahar et al. [93] reported that the fracture mode of this material comprised the interlamellar and translamellar as well as the quasi-cleavage fracture as depicted in Figure 8. However, the fatigue fracture surface was very rough owing to the large lamellar colonies in the microstructure. On the other hand, fracture in the NL occurred mostly by translamellar mode. However, for a high-Nb containing γ-TiAl alloy with NL microstructure at 750 °C, Yu et al. [98] found that the translamellar fracture mechanism transformed to translamellar and interlamellar, and then to intergranular fracture mode as shown in Figure 9.

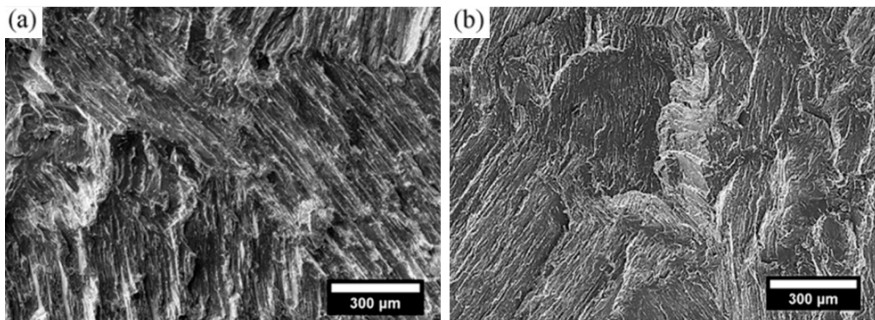

**Figure 8.** Fracture modes in FL microstructure: (**a**) Interlamellar and (**b**) quasi-cleavage fractures. SEM micrographs were reproduced with permission from Elsevier [93].

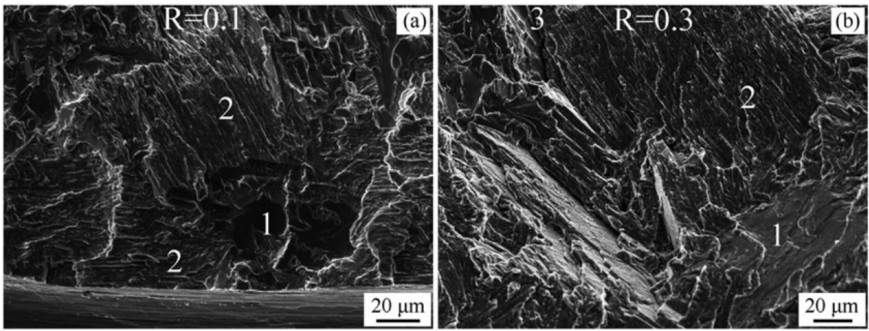

**Figure 9.** Fractographs of NL microstructure tested at 750 °C: (**a**) R = 0.1 and (**b**) R = 0.3. Features marked 1—interlamellar, 2—translamellar and 3—intergranular fractures. SEM micrographs were reproduced with permission from Elsevier [98].

According to [82], the fracture surfaces of both duplex and nearly lamellar microstructures exhibit brittle transgranular failure features of lamellar colonies. For the structures with lamellae oriented perpendicular to the loading axis, interlamellar failure is predominant, whereas for those whose orientations are along the loading direction, the failure mode is translamellar. In equiaxed γ-grains microstructures, the intergranular fracture mode dominates when the structures contain larger clusters of γ-grains in γ–γ grain boundaries; otherwise, a brittle transgranular cleavage becomes the failure mode. These fatigue crack initiation, propagation and fracture mechanisms in different microstructures can be summarised as shown in Table 4.

**Table 4.** Fatigue stage mechanisms in γ-TiAl-based alloys.

| Microstructure | Initiation | Propagation | Fracture |
|---|---|---|---|
| Duplex | Damaged surface, pores, oxide inclusions, debonded γ-grains, grain clusters, large single-phase γ-grains (by intrusion and extrusion), inter-dendritic γ-areas. | Stable crack growth (striations present when lamellae and γ-grain size < critical crack size) | Brittle transgranular cleavage or intergranular |
| Lamellar | Pores, cavities, damaged surface, oxides inclusions, interlamellar (for lamellar colonies oriented perpendicular to the load axis. | Interlamellar and translamellar cracking (fast crack growth for lamellae oriented perpendicular to the load axis) | Delamination, translamellar, stepwise fracture, quasi-cleavage fracture |

*3.2. Influence of Temperature and Environment on LCF Behaviour*

Just like in any other metals, the fatigue behaviour of γ-TiAl-based alloys is affected by temperature and the fatigue process becomes more complex at elevated temperatures. This happens because of different processes that become important at high temperatures. Some of these processes are creep, oxidation phase instabilities and dynamic strain ageing (DSA). These processes can work together or separately to make materials more prone to fatigue and reduce their lifespan [99]. It appears that the highest temperature at which titanium aluminides can function for a long time depends on how well they resist oxidation, rather than how they retain creep or strength [100]. These alloys are limited to low and intermediate temperature applications due to their insufficient oxidation resistance above 800 °C. Pilone and Felli [101] reported that when TiAl-based alloys were exposed to oxidation in air, the scale became a mixture of $Al_2O_3$ and $TiO_2$, instead of the protective alumina layer ($Al_2O_3$). Mathabathe et al. [58] studied the oxidation resistance of Ti-48Al-2Nb-0.7Cr in the air at 900 °C and observed very porous, spider web-type shape $Al_2O_3$ particles, as shown in Figure 10.

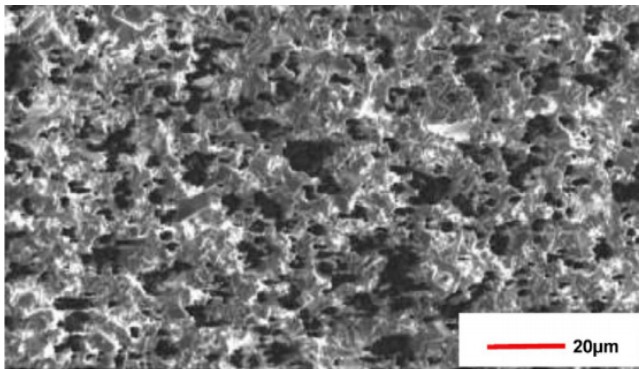

**Figure 10.** SEM image of $Al_2O_3$ scale obtained from Ti-48Al-2Nb-0.7Cr. The micrograph was reproduced with permission from Elsevier [58].

This scale is brittle and can act as an initiation site for fatigue cracks upon loading [102]. However, the oxidation resistance of TiAl-based alloys has been improved by the addition of Mo, W and Nb. Furthermore, this doping effect suppresses the formation of rutile ($TiO_2$), which is porous and allows the inward diffusion of oxygen [101].

Apart from oxidation, the fatigue crack propagation rate is found to be strongly affected by temperature. Feng et al. [103] noted the increase in the fatigue crack growth rate

in the temperature range of 25–750 °C. Nevertheless, when the temperature was between 750 and 850 °C, the rate decreased. This was attributed to the occurrence of the brittle-ductile transition of the TiAl-based alloy. Moreover, research has shown that the cyclic stress–strain response of these alloys changes at a temperature of about 650 °C. According to a study conducted by Christ et al. [104] on an investment cast alloy with a nominal composition of Ti–47Al–2Nb–2Mn (at.%) and contained 0.8 vol.%TiB$_2$, it was observed that below the temperature of 650 °C the alloy cyclically hardened, whereas a stabilised condition was quickly established at higher temperatures. This is in agreement with the cyclic stress–strain behaviour characterised by a rapid saturation of the stress amplitude, regardless of the applied strain amplitude that was observed in Ti-48Al-2Cr-2Nb (at.%) at 750 °C by Gloanec et al. [105].

Concerning cyclic deformation, the deformation mechanisms of these alloys also seem to be influenced by the temperature. Gloanec et al. [105] reported that at room temperature the deformation mechanisms of cast nearly fully lamellar Ti-48Al-2Cr -2Nb alloy are governed by the amount of strain applied. At a low total strain amplitude range, $\Delta\varepsilon_t$, the cyclic stress–strain behaviour is characterised by a hardly noticeable initial hardening followed by a stabilisation of the stress amplitude, and the deformation microstructure is only constituted of ordinary dislocations which tangle to form dipoles [105]. At intermediate $\Delta\varepsilon_t$ where the cyclic strain hardening is moderate, the formation of a vein-like structure (Figure 11a) is observed while at high $\Delta\varepsilon_t$ values a pronounced hardening is observed which is related to twinning which develops in the early stage of the fatigue life and prevents the formation of a vein-like structure. In contrast, at elevated temperatures, the deformation structure analysis suggests a high activation of dislocation climb which was confirmed by in situ TEM ageing showing shrinkage of prismatic loops [105]. However, for a high Nb-containing alloy with a lamellar microstructure, Kruml and Obrtlík [106] observed no cyclic hardening and twinning was rare at room temperature. This was attributed to the influence of Nb in reducing the twinning activity by alternating the stacking fault energy. At 750 °C, there was stable cycle behaviour with a high density of dislocation and prismatic loops (Figure 11b). Nevertheless, the alloy response changed between 750 and 800 °C, and at 800 °C, the researchers observed cyclic softening of the alloy with low dislocation density which led to the prominent destruction of lamellar colonies. This cyclic softening is attributed to the strain-induced phase transformations and dynamic recrystallization of the microstructure which occur during the high-temperature LCF process [107]. Ding et al. [108] studied the strain distribution in the microstructure of a high Nb-containing alloy subjected to a LCF loading by employing in situ and ex situ synchrotron-based high-energy X-ray at 900 °C. Their results revealed that the distribution of strain among $\alpha_2$, $\gamma$ and $\omega_o$ was different. In $\alpha_2$, the lattice strain was alternating from tensile to compressive, whereas in $\gamma$ and $\omega_o$ the lattice strain was always compressive.

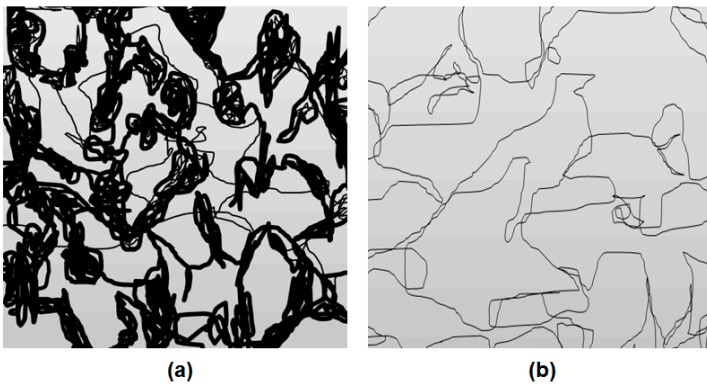

(a)                (b)

**Figure 11.** Schematic diagrams to represent (**a**) vein-like structure and (**b**) dislocation and prismatic loops.

In terms of the environment, the cyclic lifetime was found to be strongly impacted by the environment. The specimens failed faster in a vacuum and at higher temperatures, while fatigue life increases in the air as illustrated in Figure 12. Similar results were also reported in [104].

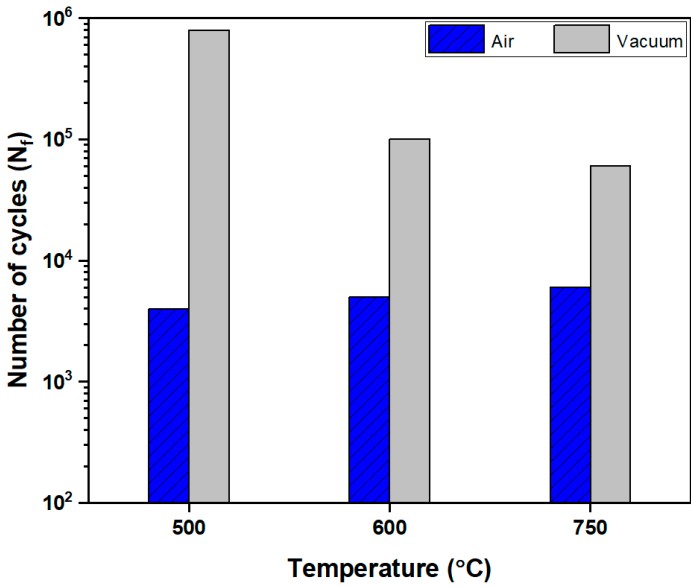

**Figure 12.** Impact of temperature on cyclic life for fatigue tests performed in air and vacuum.

The behaviour observed in Figure 12 was attributed to what Maier et al. [109] reported. Environmental effects are more pronounced at temperatures below the brittle–ductile transition temperature (BDTT), resulting in the unusual inverse effect of temperature on fatigue life for tests conducted in air environments [109].

### 3.3. LCF Life of $\gamma$-TiAl-Based Alloys

When a $\gamma$-TiAl-based alloy or structure is repeatedly stressed or strained, it will eventually break. The fatigue life of the material or structure is the number of times it can be stressed (strained) until failure occurs [110]. This number is a function of many variables, viz. how much stress (stress level), what kind of stress (stress state), how the stress changes over time (cyclic waveform), fatigue environment, and the metallurgical condition of the material. However, for $\gamma$-TiAl alloys, it has been found that the microstructure plays a major role in LCF resistance. The temperature rise or the hold time effect seems less significant than the microstructure [111]. In the literature, a general comparison is made between duplex and lamellar microstructures in terms of their fatigue life. In lamellar microstructure, cracks initiate early but propagate slowly because of crack wake ligaments. So, most of the fatigue life is spent at the crack propagation stage. Duplex microstructures, on the other hand, spend the majority of their fatigue lives nucleating micro-cracks which propagate rapidly [112]. However, in terms of the one with a better high-temperature LCF property, Recina et al. [82] reported that a duplex fine-grained structure of Ti–48Al–2W–05Si alloy had superior LCF resistance with a smaller scatter in life (shown in Figure 13) in comparison with a coarse-grained lamellar structure consisting of alternate laths of $\gamma$ and $\alpha_2$ phases. The amount of inelastic strain that occurs in each fatigue cycle affects how long the material lasts. The duplex material lasts longer because it has more isotropic hardening and less Bauschinger effect, which reduce the inelastic strains and damage in every cycle [82]. Malakondaiah and Nicholas [97] studied the high-temperature LCF life of Ti-46Al-2Nb-2Cr alloy in fully lamellar (FL) and nearly lamellar (NL) microstructural conditions at 650 and 800 °C in laboratory air with and without hold times. The tests were fully reversed (R = −1) and strain controlled (LCF) with a triangular waveform of 0.167 Hz frequency. The strain amplitude ranged from ±0.25 to ±0.45 pct. The fatigue

life data for tests at 650 and 800 °C with no hold time, in terms of reversals to failure, were plotted against the inelastic strain amplitude. Furthermore, the total strain range values were also plotted against cycles to failure for tests, with and without hold time, at 650 °C in the FL condition. The researchers observed that the alloy in the FL condition exhibited the well-known Manson–Coffin behaviour at both temperatures as evidenced by the linearity of data, whereas for the NL condition, it was inconclusive due to limited data [97]. Moreover, at 650 °C as well as at 800 °C the alloy in the NL condition possessed superior fatigue resistance as compared to the FL condition, although the difference was marginal at 800 °C [97]. Concerning the effect of hold time on fatigue life, it was observed that at 650 °C, the fatigue life of the alloy in the FL condition was doubled by a 100 s hold at peak tensile strain. However, a 100 s hold at peak compressive strain or both tensile and compressive strain reduced the fatigue life. Mean stress was the main factor that influenced the hold time effects observed [97].

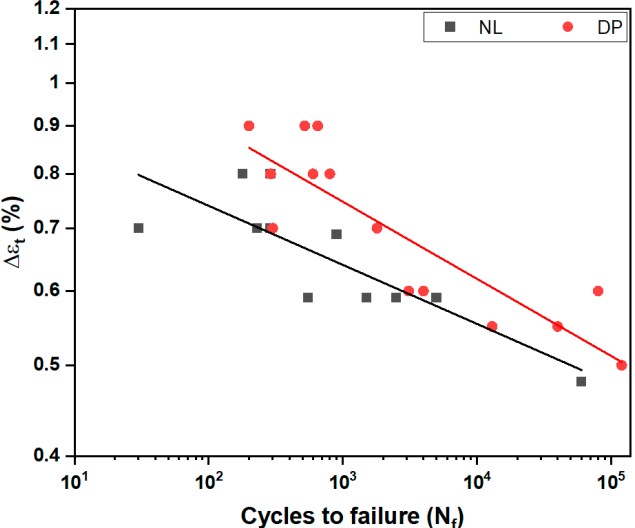

**Figure 13.** LCF life of DP and NL microstructures at 600 °C in laboratory air. Data were redrawn with permission from Elsevier [82].

The fatigue life of $\gamma$-TiAl-based alloys that contain high Nb has also been assessed in the low-cycle domain. Kruml and Obrtlík [106] systematically analysed the LCF behaviour of a TiAl-8 at.%Nb alloy with fine fully lamellar microstructure at room temperature (RT) and 700, 750 and 800 °C. The fatigue tests were performed at different strain amplitudes which were kept constant during each test. The researchers reported that the alloy exhibited the shortest fatigue life at RT when the strain amplitude was high. This was attributed to the brittle nature of the material at room temperature. However, at lower strain amplitude, the shortest fatigue life was observed to emanate from the specimens cycled at 800 °C. In Figure 14, a comparison of LCF life behaviour of Ti-48Al-2Nb-2Cr (4822) and Ti-48Al-8Nb (TiAl-8) alloys with fully lamellar microstructures is made based on the results reported in the literature [105,106]. Both alloys were cycled at a strain amplitude of ±0.4% and a strain ratio $R_\varepsilon = \varepsilon_{min}/\varepsilon_{max} = -1$ at room and elevated temperatures (25 and 750 °C). As can be seen in Figure 14, higher LCF life is displayed by the 4822 alloy at both temperatures, even though the difference is marginal at 750 °C. Nevertheless, TiAl-8 alloy is more superior in fatigue strength as evidenced by higher stress values at both temperatures. The low-cycle fatigue lives ($N_f$) of selected $\gamma$-TiAl based alloys with different microstructures at different total strain amplitudes and temperatures are illustrated in Table 5.

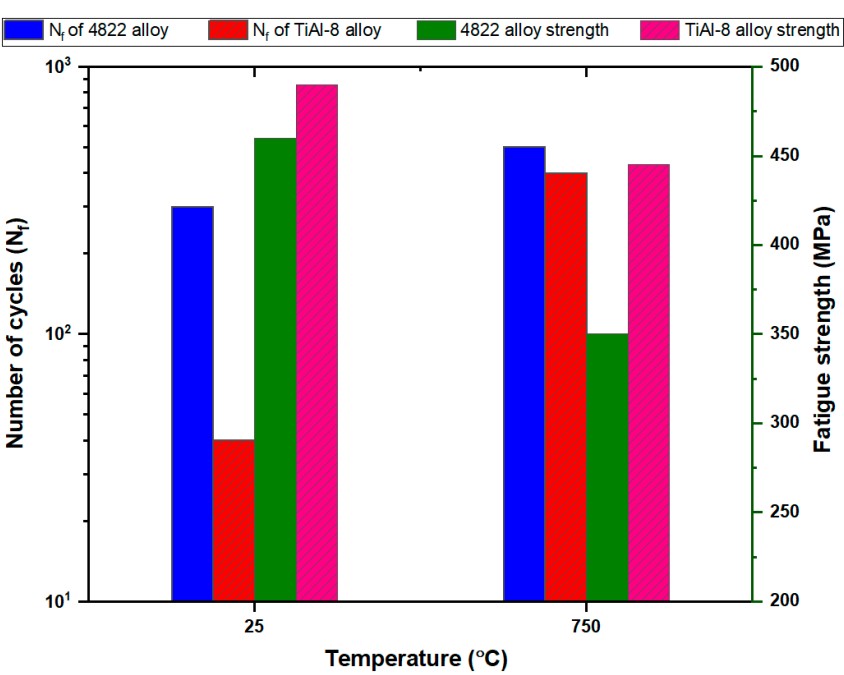

**Figure 14.** Comparison of LCF life behaviour between 4822 and TiAl-8 alloys.

**Table 5.** Low-cycle fatigue lives (Nf) of selected γ-TiAl-based alloys with different microstructures (duplex, DP; nearly lamellar, NL; nearly gamma, NG; and fully lamellar, FL) at different total strain amplitudes ($\Delta\varepsilon_t$) and temperatures.

| Alloy | Microstructure | Processing Route | $\Delta\varepsilon_t$ (%) | Temperature (°C) | $N_f$ | Ref. |
|---|---|---|---|---|---|---|
| Ti-48Al-2W-0.5Si | DP<br>NL | Casting<br>Casting | 0.6<br>0.6 | 600<br>600 | 3000<br>600 | [82] |
| Ti-48Al-2Cr-2Nb | NL<br>NG<br>NL | Casting<br>PM<br>Casting | 0.6<br>0.6<br>0.6 | 25<br>25<br>750 | 100<br>20<br>40 | [113] |
| TiAl-8 at.%Nb | FL<br>FL<br>FL | IM<br>IM<br>IM | 0.4<br>0.4<br>0.4 | 700<br>750<br>800 | 300<br>200<br>100 | [106] |
| Ti-45Al-8.5Nb-0.2W-0.2B-0.02Y | FL<br>FL<br>FL | IM<br>IM<br>IM | 0.39<br>0.3<br>0.25 | 850<br>850<br>850 | 70<br>139<br>12,075 | [92] |
| TNB-V5 (Ti-45Al-5Nb-0.2C-0.2B) | NG<br>NG<br>NG<br>NG<br>NG | IM + Extrusion<br>IM + Extrusion<br>IM + Extrusion<br>IM + Extrusion<br>IM + Extrusion | 0.575<br>0.575<br>0.575<br>0.65<br>0.65 | 400<br>600<br>800<br>400<br>800 | 133<br>94<br>641<br>86<br>206 | [114] |

PM = powder metallurgy, IM = ingot metallurgy.

## 4. Models for Low-Cycle Fatigue Behaviour of γ-TiAl-Based Intermetallic Alloys

It is a well-known fact that fatigue experiments are costly and time-consuming. Therefore, the use of models becomes essential to complement the experimental fatigue tests in predicting the low-cycle fatigue performance of γ-TiAl-based alloys. Several models exist ranging from empirical to theoretical models that can be used to assess the fatigue property of the alloys in the low-cycle regime. However, in this review, only models that have been assessed for the prediction of fatigue crack initiation and propagation lives as well as the total fatigue life are discussed.

### 4.1. Fatigue Crack Initiation Models

Despite the existence of several crack initiation criteria which are employed in other engineering materials such as 304L austenitic stainless steel, as elucidated in [115] and

other models discussed elsewhere [116], to the best of our knowledge, no information in the literature is found to indicate the application of the criteria to $\gamma$-TiAl-based alloys. However, Feng et al. [117] assessed the accuracy of the newly proposed fatigue crack initiation method based on qualitative and quantitative approaches. The model estimates the total interaction forces ($F_I$ and $F_{II}$) acting at dislocations I and II from the domain antiphase boundary (APB) to be given by Equations (1) and (2), respectively [117]:

$$F_I = -\frac{\mu b^2}{2\pi(12-v)w} + \frac{\mu b^2 d(d^2-h^2)}{2\pi(12-v)(d^2+h^2)} + \frac{\mu^2 b^2 (d+w)\left[(d+w)^2-h^2\right]}{2\pi(12-v)\left[(d+w)^2+h^2\right]^2} + \gamma_{APB} \quad (1)$$

$$F_{II} = \frac{\mu b^2}{2\pi(12-v)w} + \frac{\mu b^2 d(d^2-h^2)}{2\pi(12-v)(d^2+h^2)} - \frac{\mu b^2 (w-d)\left[(w-d)^2-h^2\right]}{2\pi(12-v)\left[(w-d)^2+h^2\right]^2} - \gamma_{APB} \quad (2)$$

where $\mu$ is the shear modulus, $v$ is Poisson's ratio, $b$ is the magnitude of the Burgers vector ($\tilde{b}$), $w$ distance between dislocations, $d$ is the misalignment of dislocations, $h$ is the height between the top and bottom dislocations, and $\gamma_{APB}$ is the APB energy. The researchers observed that the application of the model to TiAl alloys yielded more accurate results. However, they pointed out that the model ignored several factors in addition to dislocations and debris that have an effect on fatigue life such as temperature, loading frequency and stress ratio. This suggests that there is a need of shifting research interests towards this area.

*4.2. Fatigue Crack Propagation Models*

When it comes to the fatigue damage tolerance design of engineering materials, the fatigue crack growth (FCG) behaviour is critical. A vast amount of research has been carried out in this area mostly by the application of the famous Paris model [116] to investigate the fatigue crack growth property of $\gamma$-TiAl-based alloys under different conditions. The Paris model based on linear elastic fracture mechanics (LEFM) is expressed in Equation (3).

$$\frac{da}{dN} = C(\Delta K)^m \quad (3)$$

where $a$ is the crack length, $N$ is the number of cycles, $da/dN$ is the fatigue crack growth rate (FCGR), $K$ is the stress intensity factor, and $\Delta K$ is the stress intensity factor range with $\Delta K = K_{max} - K_{min}$, and $C$ and m are material constants. Graphically, the model is represented as exhibited in Figure 15.

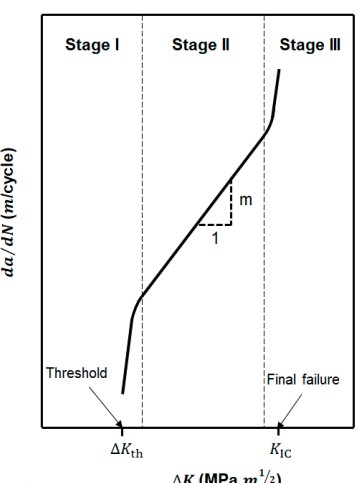

**Figure 15.** Schematic representation of FCG curve.

As shown in Figure 15, there are three stages and the Paris law is successfully applied in the central region (stage II) to describe FCGR for specific values of stress ratio. The characteristics of the three stages in LCF have already been discussed earlier and summarised in Table 4. Therefore, the main focus of this section is to elucidate the responses of different $\gamma$-TiAl-based alloys to the application of the Paris model. Dahar et al. [93] investigated the effects of applied load ratio, R, in different specimen orientations on the room temperature FCGR on as-cast Ti-48Al-2Nb-2Cr alloy with a fully lamellar microstructure. They observed that the FCGR strongly depended on the load ratio as illustrated in Figure 16. The larger the load ratio, the higher the crack growth rate and the significantly smaller the fatigue threshold $\Delta K_{th}$, whereas the $K_{IC}$ at failure was not significantly affected by R.

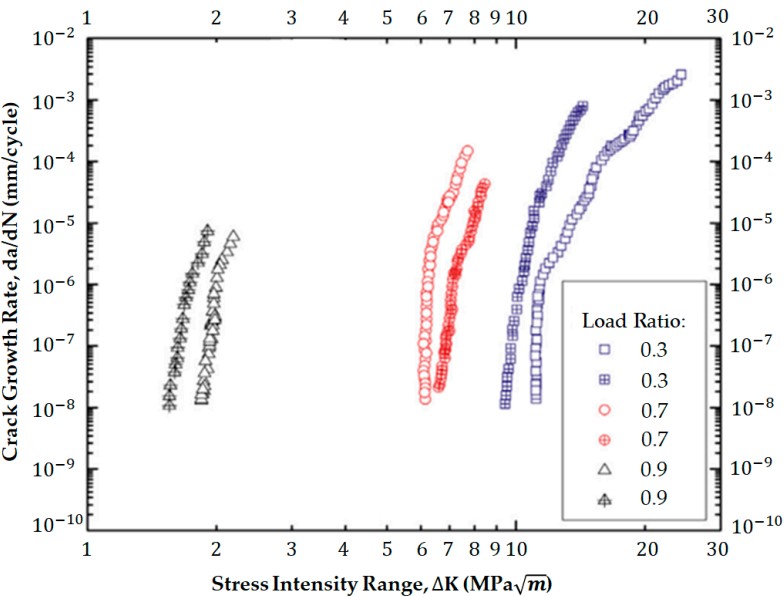

**Figure 16.** Load ratio influence on FCGR at the temperature and frequency of 25 °C and 20 Hz, respectively. Reproduced with permission from Elsevier [93].

In a different study conducted by Trant et al. [94], the FCGR of a cast and hot isostatic pressed (HIP'ed) 4522XD $\gamma$-TiAl alloy with lamellar microstructure at different temperatures between 400 and 750 °C was examined. They found that both the FCGR and $\Delta K_{th}$ were higher at 750 °C than at 400 °C as demonstrated in Figure 17. However, the researchers noted that heating and cooling of specimen between the temperatures of 400 and 750 °C during fatigue testing resulted in retardation of the crack growth rate. Concerning the FCGR dependency on lamellar orientation, Yang et al. [118] reported that in a cast Ti-46Al-8Nb alloy with coarse lamellar microstructure and at a temperature of 650 °C, the lamellar colonies with their interfaces parallel to the stress axis had the highest fatigue crack growth threshold. On the other hand, the ones that were oriented at the range of 40–65° from the stress axis were found to possess the lowest FCG threshold. Gloanec et al. [119] also investigated the FCG behaviour of a Ti-48Al-2Cr-2Nb alloy which was fabricated by casting (in fully lamellar) and powder metallurgy (in nearly $\gamma$). The applied load ratio range was 0.1–0.5 and the experiment was carried out in laboratory air at room temperature. The researchers observed that the lower FCGR emanated from the cast alloy with the fully lamellar microstructure at R = 0.1. Moreover, they reported that the crack closure effects were more profound in the cast alloy than in the one prepared by powder metallurgy (PM). Furthermore, they pointed out that when the load ratio was increased above 0.4, no crack closure was detected in the PM alloy, whereas for the cast alloy a more load ratio (i.e., above R = 0.45) was needed to eliminate the closure effect.

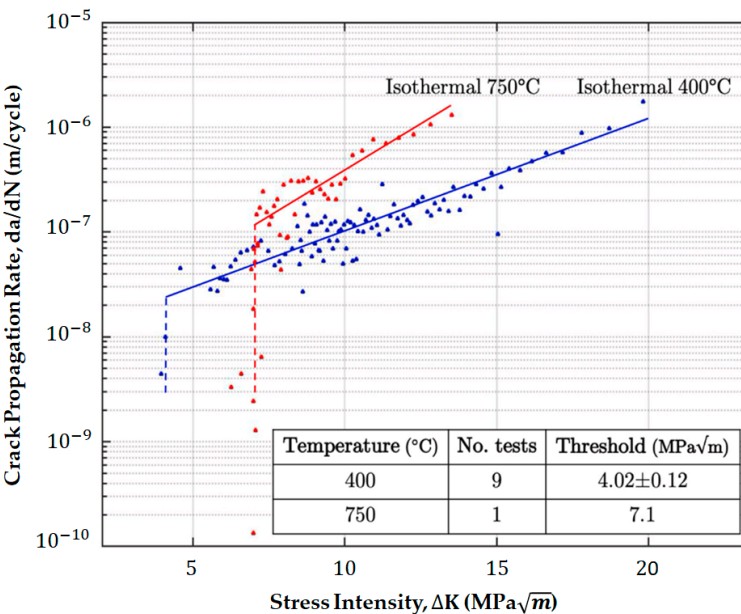

**Figure 17.** Isothermal FCG curves of as-cast and HIP 4522XD alloy at 400 and 750 °C. Reproduced with permission from Elsevier [94].

It is noteworthy that the Paris model (Equation (3)) has been modified by several researchers to incorporate other important factors that significantly influence the FCGR of these intermetallic alloys. For instance, to reflect the influence of average stress on the crack propagation rate and to show the effects of accelerated crack propagation when the stress intensity factor $\Delta K$ approaches the critical value $K_{IC}$, Forman [103] modified the Paris model as follows:

$$\frac{da}{dN} = \frac{C(\Delta K)^m}{(1-R)K_{IC} - \Delta K} \tag{4}$$

where $C$ and $m$ are the material constants, $K_{IC}$ is the critical value of the stress intensity factor and $R$ is the stress ratio. This model was designed to describe stages II and III for medium and rapid fatigue crack propagation, respectively. To take into account both stage I and II of crack propagation, Zheng and Hirt [103] also modified the Paris model to produce a model known as the passivation cracking model of the crack tip which is expressed in Equation (5).

$$\frac{da}{dN} = B(\Delta K - \Delta K_{th})^2 \tag{5}$$

where $B$ and $\Delta K_{th}$ represent the fatigue crack growth factor and the threshold value, respectively. Furthermore, to demonstrate the influence of temperature on fatigue crack propagation of stage II, Feng et al. [103] amalgamated models (3), (4) and (5) to design a new model expressed in Equation (6).

$$\frac{da}{dN} = C(T)\left(\Delta K_{eff}\right)^m \tag{6}$$

where $T$ is the temperature, $C(T)$ is the temperature dependent coefficient and $\Delta K_{eff}$ is the variable range of the effective stress intensity factor. In general, and as elucidated by Edwards et al. [87] the Paris slope, $m$ (Figure 15) ranges from 7 to 50 for TiAl alloys. This depends on the condition of the microstructure, and above 750 °C, the slope decreases significantly as the materials transition from brittle to ductile states. Table 6 exhibits the values of $m$, fatigue threshold ($\Delta K_{th}$) and overload failure ($K_{IC}$) of various $\gamma$-TiAl-based alloys with different microstructures fabricated by various processing routes. The alloys were tested mainly at room temperature and different load ratios ($R$).

**Table 6.** Values of fatigue crack growth (FCG) parameters for selected γ-TiAl-based alloys with different microstructures.

| Alloy | Microstructure | Processing Route | Temperature (°C) | R | $\Delta K_{th}$ | m | $K_{IC}$ | Ref. |
|---|---|---|---|---|---|---|---|---|
| Ti-48Al-2Nb-2Cr | FL | Casting | 25 | 0.1 | 9.2 | 6.5–10.4 | 20.4 | [93, 119] |
| | FL | Casting | 25 | 0.3 | 8 | 23.2 | 18.3 | |
| | FL | Casting | 25 | 0.7 | 5.8 | 35.8 | 24.6 | |
| | FL | Casting | 25 | 0.9 | 2.2 | 78.1 | 24.1 | |
| | Equiaxed γ-grains | PM | 25 | 0.1 | 5 | 8.39 | 8 | |
| | NL | Casting | 25 | 0.1 | 7.5 | 8.11 | 30 | |
| | NL | Casting | 25 | 0.5 | - | 8.11 | - | |
| 4522XD | FL | Cast + HIPed | 400 | 0.1 | 4 | - | - | [94] |
| | FL | Cast + HIPed | 750 | 0.1 | 7.1 | - | - | |
| Ti-43.5Al-4Nb-1Mo-0.1B (TNM) | NL | Casting (T-L) | 25 | 0.1 | 9.9 | 9 | 24 | [120] |
| | NL | Cast + HIPed | 25 | 0.1 | 5.9 | 6 | 19.4 | |
| | DP | Cast + Forging | 25 | 0.1 | 9.3 | 14 | 18.3 | |
| | NL | Cating (T-L) | 25 | 0.7 | 4.8 | 50 | 18.8 | |
| | NL | Cast + HIPed | 25 | 0.3 | 4.1 | 9.3 | 17.2 | |
| | DP | Cast + Forging | 25 | 0.7 | 3.7 | 35 | 15.3 | |

FL = fully lamellar; NL = nearly lamellar; DP = duplex; T-L = casting direction.

### 4.3. Fatigue Life Models

As already defined in Section 3.3, the fatigue life is the number of stress/strain cycles a component can withstand before failure takes place [116]. In engineering applications where the loading is predominantly cyclic, it is used as an indicator for the prediction of components' durability and reliability. Several models can be employed to predict the fatigue life of γ-TiAl components. However, in LCF, in which the applied stresses exceed the yield stress and reach a plastic region, the common approach is the use of an empirical model known as the Coffin–Mason model [116] (strain–life model) which is obtained by fitting a large amount of data. The Coffin–Mason model is given as indicated in Equation (7).

$$\varepsilon_a = \frac{\Delta\varepsilon_e}{2} + \frac{\Delta\varepsilon_p}{2} = \frac{\sigma'_f}{E}\left(2N_f\right)^b + \varepsilon'_f\left(2N_f\right)^c \tag{7}$$

where $\varepsilon_a$ is the total strain; $\Delta\varepsilon_e$ is the elastic strain amplitude; $\Delta\varepsilon_p$ is the plastic strain amplitude; $\sigma'_f$ and $\varepsilon'_f$ are the fatigue strength and ductility coefficients, respectively; $b$ and $c$ are the fatigue strength and ductility exponents, respectively; and $N_f$ is the fatigue life and $E$ is the elastic modulus. The Coffin–Mason Equation (7) is often represented graphically by a strain versus the number of cycles curve popularly referred to as an $\varepsilon$-$N_f$ curve with its axes plotted in a logarithmic scale as shown in Figure 18. To tackle fatigue problems, the Coffin–Mason model is also often amalgamated with the linear cumulative damage model (Miner's rule algorithm) [116], as illustrated in Figure 18. The Miner's damage accumulation rule is given by Equation (8):

$$D_{tot} = \sum_{i=1}^{n} D_i = \sum_{i=1}^{n} \frac{1}{N_{fi}} \tag{8}$$

where $N_f$ is the total number of cycles to failure, and $n$ is the number of strain levels in the block loading spectrum.

As already discussed in Section 3.3, several researchers have applied the model to different γ-TiAl-based alloy systems. Recina et al. [82] fitted the fatigue life data of Ti-48Al-2W-0.5Si alloy with both duplex fine-grained and coarse-grained lamellar structures. The researchers observed that both data sets fitted well on the Coffin–Mason model. Furthermore, Malakondaiah and Nicholas [97] also reported that the FL microstructure of Ti-46Al-2Nb-2Cr alloy exhibited the Manson–Coffin behaviour at elevated temperatures. Apart from the empirical models, energy-based models have also been assessed in γ-TiAl alloys. These models mainly aim at analysing the nonlinear deformation behaviour in the fatigue regime. Gloanec et al. [113] determined the adequacy of two energy-based criteria to assess low-cycle fatigue lifetime for a quaternary alloy Ti-48Al-2Cr-2Nb (at.%) with

lamellar and fine equiaxed γ-grains microstructures at temperatures of 25 and 750 °C. The first fatigue criterion tested was proposed by Constantinescu et al. [121], which is based on the dissipated energy per cycle and expressed in Equation (9):

$$D = \alpha N_f^\beta \tag{9}$$

where α and β are the fatigue criterion coefficients; and D is the dissipated energy and equates to the surface of the hysteresis loop in the strain–stress curve. The second fatigue criterion corrects the dissipated energy with a hydrostatic pressure term and is expressed in Equation (10):

$$D + cP_{max} = \alpha N_f^\beta \tag{10}$$

where $P_{max}$ is the maximal hydrostatic pressure reached during the stabilised cycle and c is a material constant. The researchers found that both criteria gave some sort of linearity. However, it was the first criterion (Equation (9)) which gave the better fatigue lifetime prediction at both temperatures. Moreover, they stated that the identification of the fatigue criterion coefficients for both models was significantly affected by the temperature, applied strain ratio and the microstructures of the alloys. In another study, Roth and Biermann [122] evaluated the correlation between the LCF lives of the 3ʳᵈ-generation γ-TiAl alloy TNB-V5 with a nominal composition of Ti–45Al–5Nb–0.2C-0.2B (at.%) and the Smith–Watson–Topper damage parameter ($P_{SWT}$). The alloy was fabricated by vacuum arc melting followed by extrusion process and its microstructure was near-gamma with lamellar grains volume fraction of approximately 10%. The LCF tests were conducted at temperatures of 400, 600 and 800 °C and a constant mechanical strain rate of $4 \times 10^{-3}$ s$^{-1}$ with applied mechanical strain amplitudes ($\Delta\varepsilon_{mech}/2$) of $5.75 \times 10^{-3}$ and $6.50 \times 10^{-3}$. The Smith–Watson–Topper damage model is expressed in Equation (11):

$$P_{SWT} = \sqrt{\sigma_{max}\varepsilon_a E} \tag{11}$$

where $\sigma_{max}$ is the maximum tensile stress, $\varepsilon_a$ is the mechanical strain amplitude ($\Delta\varepsilon_{mech}/2$) and E is the Young's modulus. The researchers found that the damage model by Smith, Watson and Topper was well-suitable for a general LCF life prediction of the alloy as exhibited in Figure 19. A summary of selected models that have been assessed for the prediction of low-cycle fatigue behaviour of γ-TiAl-based alloys is displayed in Table 7.

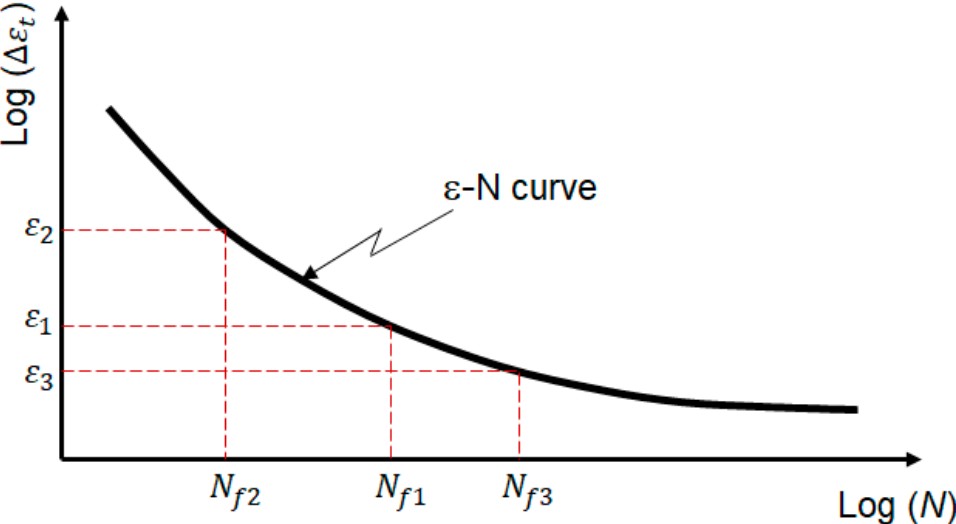

**Figure 18.** Schematic representation of ε-N curve.

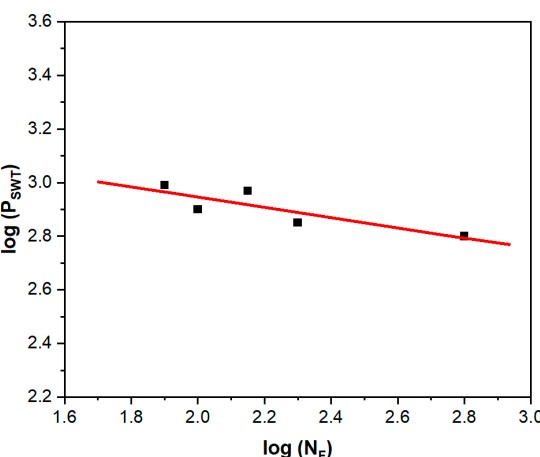

**Figure 19.** Damage parameter ($P_{SWT}$) vs. fatigue life ($N_F$) plot in a logarithmic plot for LCF tests of TNB-V5 alloy. Data were redrawn with permission from Elsevier [122].

**Table 7.** Assessed models for the prediction of LCF behaviour of γ-TiAl-based alloys and their remarks.

| Classification | Designation | Expression | Remarks |
|---|---|---|---|
| Fatigue crack initiation | Total interaction forces model | $F_I = -\frac{\mu b^2}{2\pi(12-v)w} + \frac{\mu b^2 d(d^2-h^2)}{2\pi(12-v)(d^2+h^2)}$ $+ \frac{\mu^2 b^2(d+w)\left[(d+w)^2-h^2\right]}{2\pi(12-v)\left[(d+w)^2+h^2\right]^2} + \gamma_{APB}$ $F_{II} = \frac{\mu b^2}{2\pi(12-v)w} + \frac{\mu b^2 d(d^2-h^2)}{2\pi(12-v)(d^2+h^2)}$ $- \frac{\mu b^2(w-d)\left[(w-d)^2-h^2\right]}{2\pi(12-v)\left[(w-d)^2+h^2\right]^2} - \gamma_{APB}$ | Yields relatively accurate results. It needs improvement. |
| Fatigue crack propagation | Paris model | $\frac{da}{dN} = C(\Delta K)^m$ | Commonly used and gives accurate results only in stage II. |
| | Forman model | $\frac{da}{dN} = \frac{C(\Delta K)^m}{(1-R)K_{IC}-\Delta K}$ | Suitable for stages II and III. |
| | Zheng–Hirt model | $\frac{da}{dN} = B(\Delta K - \Delta K_{th})^2$ | Suitable for stages I and II. |
| | Feng model | $\frac{da}{dN} = C(T)(\Delta K_{eff})^m$ | Characterises only stage II. A new model is needed for all the stages. |
| Fatigue life | Strain–life model | $\varepsilon_a = \frac{\sigma_f'}{E}(2N_f)^b + \varepsilon_f'(2N_f)^c$ | Gives accurate results. |
| | Energy-based model | $D = \alpha N_f^\beta$ | Suitable for LCF life prediction. |
| | | $D + cP_{max} = \alpha N_f^\beta$ | Not suitable for LCF life prediction. |
| | Smith–Watson–Topper damage model | $P_{SWT} = \sqrt{\sigma_{max}\varepsilon_a E}$ | Suitable for accurate LCF life prediction of γ-TiAl alloys. |

## 5. Microstructural Optimisation for Improved LCF Life of γ-TiAl-Based Alloys

The microstructures of $\gamma/\alpha_2$ titanium aluminides can be optimised by thermo-mechanical processing and/or subsequent heat treatments [123,124]. Crofts et al. [125] listed a set of parameters, viz. colony size, lamellar thickness, proportion of $\alpha_2$, serration of the grain boundaries and nature of the lamellar interfaces that can be altered or varied through heat treatments or thermo-mechanical processing. However, the role of the microstructure and the characteristic dimensions of the lamellar colonies emerge as the fundamental parameter to determine fatigue strength [89]. Recina et al. [89] pointed out that apart from minimum levels of defects in the microstructure and the path of solidification, an ideal structure for optimised low-cycle fatigue performance in the alloy (Ti-48Al-2W-0.5Si) must be a duplex one obtained with a small lamellar colony size, where the size of the equiaxed γ-grains is below 100 μm and where the interdendritic γ areas are minimised. On the other hand, other studies indicate that a fine-grained, fully lamellar structure is beneficial for strength and fatigue resistance in combination with high fracture toughness [126]. Furthermore, Chan and Kim [127] reported that the smaller the space between the lamellae in TiAl alloys,

the harder it is to start and propagate a crack. However, the colony size also affects both initiation and crack growth toughness in a more complicated way. The space between the lamellae determines how much microcracking occurs across the lamellae and how big the shear ligaments are. When the space is small, microcracking across the lamellae is less likely, and the main crack has more difficulty connecting with microcracks between the lamellae. This leads to bigger shear ligaments and more resistance to cracking. When the space is large, microcracking across the lamellae is more likely, and the main crack can easily connect with microcracks across and between the lamellae. This leads to smaller shear ligaments and less resistance to cracking [127].

From the above discussion, it appears that the key to obtaining improved mechanical properties (fatigue as well as other properties) in TiAl alloys is to use a processing route that will ensure grain refinement and texture-free microstructures. As stated at the beginning of this subsection, different processing routes can be employed to engineer microstructure [120]. Several research groups investigated how to improve the mechanical properties of $\beta$-solidifying $\gamma$-TiAl-based alloys by developing different processes and influencing their microstructural evolution. They mainly used hot-forming operations and heat treatments to achieve the desired microstructures, starting from either a cast/HIP ingot or a powder metallurgical pre-material [24,35,128]. In these investigations, forming procedures, such as hot-extrusion, hot-rolling and hot-forging, are playing a central role in the evolution of the final microstructure. During these thermo-mechanical processes, an increased dislocation density can help to refine coarse microstructural constituents through recrystallization processes [128]. By applying different heat treatments in a sequence, the microstructure can be changed to achieve optimal mechanical properties for the material after the final heat treatment, as shown in [129]. Recently, Burtscher et al. [130] have reported interesting results that have indicated a significant improvement in the strength of a cast/hot iso-static pressed and hot-extruded TNM Ti–43.5Al–4Nb–1Mo–0.1B (at.%) alloy. In the study, the researchers obtained four different degrees of deformation ($\varphi$), viz.: 0, 0.6, 1.4 and 1.9. After the samples were hot-extruded to these ratios, a one-step heat treatment (the parameters were not disclosed) was conducted on each set of the samples to achieve a stable microstructure close to thermodynamic equilibrium [130]. The fine-grained and homogenous microstructure with a globular $\gamma$ mean grain diameter of 1.8 μm was attained at the deformation of 1.4. After employing room temperature high-cycle fatigue and tensile tests to the resulting extruded alloy, a fatigue strength of 950 MPa, a fracture strain of about 1.3% and yield strength of 1121 MPa were obtained. Moreover, at 800 °C, the yield strength amounted to 404 MPa and a fracture strain of more than 87% could be obtained. Additionally, decreasing grain size was observed, which resulted in a significantly lower brittle-to-ductile transition temperature [130]. However, even though the LCF tests were not performed in this study, it is clear that the effect of grain refinement and homogenization during the deformation process provides promising and balanced mechanical properties both at ambient and elevated temperatures [130].

## 6. Summary

$\gamma$-TiAl-based alloys consisting of majorly $\gamma$-phase and 5–20% phase volume of $\alpha_2$ have engineering significance. However, the alloys in their binary TiAl form are inherently brittle and possess inadequate property values to meet the requirements of any engine components. Therefore, several alloying elements such as Nb, Mn, Cr, V, Mo, B, C, Si, Ta, Zr, Sn, Fe and Y are added to the TiAl to improve their thermal and mechanical properties. This has resulted in the designing and developing of several $\gamma$-TiAl-based alloy systems which have been implemented in automobile and aircraft engines. In service, the components are inevitably subjected to cyclic stresses or strains. The crack initiation mechanisms in the LCF regime mainly depend on whether the microstructure is of duplex or lamellar type. For the duplex microstructure, crack initiation occurs either at weak spots, such as surface damage, pores and oxide inclusions but also at debonded $\gamma$-grains or grain clusters acting as stress raisers, or in larger single-phase $\gamma$-grains. Conversely, crack initiation in the

lamellar microstructures takes place in the interlamellar regions in addition to the surface damage, pores and oxide inclusions, and is facilitated by the lamellae oriented with the lamellar laths perpendicular to the loading axis. The mechanisms for crack propagation in lamellar microstructures include interlamellar and translamellar cracking with a tortuous crack path for lamellar laths oriented perpendicular to the crack direction, whereas, for the ones oriented parallel to the crack direction, the mode is characterised by fast crack growth. On the other hand, stable crack growth is observed in duplex microstructure with the presence of striations when lamellae and $\gamma$-grain sizes are less than the critical crack size. The fracture mechanisms in lamellar microstructures involve delamination, translamellar, stepwise and quasi-cleavage fractures while in duplex microstructures, the mechanisms consist of brittle transgranular cleavage or intergranular fractures.

LCF life and deformation mechanisms are strongly influenced by temperature and environment. In the air, the fatigue life increases with temperature, and it is attributed to the crack closure effect induced by oxides at low load ratios. Conversely, in a vacuum, the fatigue life decreases with increasing temperature. At lower temperatures, the cyclic deformation mechanisms of low Nb-containing alloys are characterised by cyclic hardening and twinning with the formation of vein-like structures as the strain amplitude is increased. However, cyclic hardening and twinning are rarely observed for high Nb-containing alloys at room temperature owing to the influence of Nb in reducing the twinning activity by alternating the stacking fault energy. At elevated temperatures, the deformation structure analysis suggests a high activation of dislocation climb with prismatic loops, followed by a cyclic softening. In general, the low Nb-containing alloys possess higher LCF life at both room and elevated temperatures. However, the high Nb-containing alloys are superior in fatigue strength.

Despite the existence of several fatigue crack initiation criteria which are employed in other engineering materials, the application of the models to predict the LCF crack initiation life of $\gamma$-TiAl-based alloys is elusive. However, a new proposed model which estimates the total interaction forces acting at dislocations from the domain antiphase boundary (APB) yielded more accurate results when applied to TiAl alloys. Concerning fatigue crack growth, the commonly applied model is the Paris law. The model has been observed to fit well with the FCG data of $\gamma$-TiAl-based alloys in which the Paris slope strongly depends on several factors, viz. microstructures, load ratio, temperature and lamellar orientation. For LCF life prediction, the Coffin–Mason, the dissipated energy per cycle, and the Smith–Watson–Topper damage models have been assessed and found well suitable for a general LCF life prediction of the alloys. To improve the LCF life of these alloys, microstructural optimisation obtained from well-designed thermo-mechanical fabrication processes is critical. These fabrication processes should ensure a minimum level of porosity, inclusions and inter-dendritic $\gamma$ areas. Moreover, they should result in small lamellar colony size with equiaxed $\gamma$-grain sizes below 100 μm for the duplex structures and fine grains with minimum lamellar spacing for the lamellar microstructures.

**Author Contributions:** J.J.M.E.: literature review and manuscript drafting; M.N.M.: literature review and editing; C.W.S. literature review and editing; A.S.B.: final manuscript review, editing and manuscript structure. All authors have read and agreed to the published version of the manuscript.

**Funding:** This research was funded by Thuthuka National Research Foundation grant no: 138314. And The APC was funded by the Council of Scientific and Industrial Research (CSIR).

**Data Availability Statement:** Data sharing not applicable.

**Acknowledgments:** The Council of Scientific and Industrial Research (CSIR), National Research Foundation and the University of Pretoria (UP) in the Republic of South Africa are acknowledged for funding this work.

**Conflicts of Interest:** The authors declare no conflict of interest.

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
