# Peer review of "Low-Cycle Fatigue Behaviour of Titanium-Aluminium-Based Intermetallic Alloys: A Short Review"

_metals, doi:10.3390/met13081491_

Round 1

Reviewer 1 Report

In this manuscript, the review on Low-cycle fatigue behaviour of Titanium-aluminium based intermetallic alloys in terms of crack initiation, propagation and fracture mechanisms, and the effect of temperature and environment on cyclic deformation mechanisms and the resulting fatigue life was provided. The comprehensive discussion about modelling and prediction of the fatigue property of these alloys with regard to the initiation and propagation lives as well as the total fatigue life has been presented. The effective methods of optimizing the microstructures of TiAl-based alloys to ensure improved LCF behaviour have been also reviewed. Despite the overall interest, the paper is needed to do major revisions before being considered for publication.

1. A lot of content in the manuscript has been presented in the textbook, so it is lack of novelty. For example, page 8, from line 172-254.

2. There are many references of earlier years in this manuscript, it is 10 years ago, so far as to be 20 years ago. And the format of some references is irregularity, for example, reference [33],[78] and [95-96] et al.

3. The results and methods of studies are presented using the large paragraphs of text, but the figures, tables and curves of test results is not provided sufficiently the manuscript.

 Suggestion:

1. Mass streamlining about content in the manuscript will be done, in special the content in textbook as known in early stage.

2. The figures, tables and curves of test results will be used to demonstrate the research status of Titanium-aluminium based intermetallic alloys in the manuscript.

3. Some relative references published recently will be replenished.

The quality of English language of this manuscript is ok.

Author Response

Q 1:

Mass streamlining of content in the manuscript will be done, in special the content in the textbook as known in early stage.

Response:

Mass streamlining was done. Furthermore, the manuscript was also assessed for similarity index. The changes made were highlighted as indicated in section 1 (pages 1-5), subsection 3.1 (pages 9, 10, 11, 13, 14) and section 5 (page 24). Gamma titanium aluminide textbooks:

  • Gamma Titanium Aluminide Alloys (2011) by Appel et al,
  • Gamma Titanium Aluminides (1999) by Kim et al,
  • Gamma Titanium Aluminides (2003) by Kim et al,
  • Gamma Titanium Aluminides (2014) by Kim et al.

Were consulted and it was discovered that the discussion on LCF mechanisms was mainly on cyclic deformation of the alloys. Therefore, a gap was found that needed to be closed by a thorough discussion on microstructural mechanisms using both recent and old studies owing to the limited number of articles.                                     

Q 2

The figures, tables and curves of test results will be used to demonstrate the research status of Titanium-aluminium based intermetallic alloys in the manuscript.

Response:

Figures 6, 7, 8, 9, 10, 13, 16, 17 and 19, and Table 7 were incorporated into the manuscript.

Q 3:

Some relative references published recently will be replenished.

Response:

Recently published references were included and highlighted in the manuscript to replace some of the earlier published ones.

Reviewer 2 Report

Dear Authors,

I have read your paper "

Low-cycle fatigue behaviour of Titanium-aluminium based intermetallic alloys: A review"scarefully.

Explanations are clear and the review is easy to read. 

However, it requires few corrections

Has the PRISMA guidelines been followed?

Please add, the methodological sections in which it is described in detail how the review has been done.

How can the reader know if the review is a complete review?

The sections content includes group citations of articles. Especially in a review article, such groupings should be avoided. More details on the quoted papers should be provided.

The paper can be accepted for publication after minor improvements.

Author Response

Q 1:

Please add the methodological sections in which it is described in detail how the review has been done.

Response:

To describe in detail how the review was done, section 2 (on pages 8 and 9) was added to discuss the employed methodology and search strategy.

Q 2:

How can the reader know if the review is a complete review?

Response:

The word “short” was added to the title to show that it is a short review.

Q 3:

The section's content includes group citations of articles. Especially in a review article, such groupings should be avoided. More details on the quoted papers should be provided

Response:

Group citations were removed and more details about the cited articles were provided.

Round 2

Reviewer 1 Report

The manuscript has been revised according to the referees’ comments by author, and the reviewer thinks it has been fit for publishing in the journal.